# The Effects of Prior Mammography Screening on the Performance of Breast Cancer Detection in Taiwan

**DOI:** 10.3390/healthcare10061037

**Published:** 2022-06-02

**Authors:** Chin-Chuan Chang, Tzu-Chuan Ho, Chih-Ying Lien, Daniel Hueng-Yuan Shen, Kuo-Pin Chuang, Hung-Pin Chan, Ming-Hui Yang, Yu-Chang Tyan

**Affiliations:** 1Department of Nuclear Medicine, Kaohsiung Medical University Hospital, Kaohsiung 807, Taiwan; chinuan@gmail.com; 2School of Medicine, Kaohsiung Medical University, Kaohsiung 807, Taiwan; 3Neuroscience Research Center, Kaohsiung Medical University, Kaohsiung 807, Taiwan; 4Department of Electrical Engineering, I-Shou University, Kaohsiung 840, Taiwan; 5Department of Medical Imaging and Radiological Sciences, Kaohsiung Medical University, Kaohsiung 807, Taiwan; tzuchuanho@gmail.com (T.-C.H.); lainamylove@gmail.com (C.-Y.L.); 6Department of Nuclear Medicine, Kaohsiung Veterans General Hospital, Kaohsiung 813, Taiwan; hyshen@vghks.gov.tw (D.H.-Y.S.); hpchan@vghks.gov.tw (H.-P.C.); 7Graduate Institute of Animal Vaccine Technology, College of Veterinary Medicine, National Pingtung University of Science and Technology, Pingtung 912, Taiwan; kpchuang@g4e.npust.edu.tw; 8Department of Medical Education and Research, Kaohsiung Veterans General Hospital, Kaohsiung 813, Taiwan; 9Center of General Education, Shu-Zen Junior College of Medicine and Management, Kaohsiung 821, Taiwan; 10Graduate Institute of Medicine, College of Medicine, Kaohsiung Medical University, Kaohsiung 807, Taiwan; 11Department of Medical Research, Kaohsiung Medical University Hospital, Kaohsiung 807, Taiwan; 12Center for Cancer Research, Kaohsiung Medical University, Kaohsiung 807, Taiwan; 13Research Center for Environmental Medicine, Kaohsiung Medical University, Kaohsiung 807, Taiwan

**Keywords:** breast cancer, mammography, cancer screening, positive predictive value, recall rate

## Abstract

The aim of this study was to investigate the influence of previous mammography screening on the performance of breast cancer detection. The screened women were divided into first-visit and follow-up groups for breast cancer screening. The positive predictive value (PPV), cancer detection rate (CDR), and recall rate were used to evaluate and analyze the overall screening performance among the two groups. Among them, 10,040 screenings (67.2%) were first visits and 4895 screenings (32.8%) were follow-up visits. The proportion of positive screening results for first-visit participants was higher than that for their follow-up counterparts (9.3% vs. 4.0%). A total of 98 participants (74 first-visit and 24 follow-up visit) were confirmed to have breast cancer. The PPV for positive mammography for women who underwent biopsy confirmation was 28.7% overall, reaching 35.8% for the follow-up visit group and 27.0% for the first-visit group. The CDR was 6.6 per 1000 overall, reaching 7.4 per 1000 for first-visit group and 4.9 per 1000 for the follow-up group. The overall recall rate was 7.9%, reaching 9.7% for the first-visit group and 4.2% for the follow-up group. The PPV is improved and the recall rate is decreased if prior mammography images are available for comparison when conducting mammography screening for breast cancer. By this study, we concluded that prior mammography plays an important role for breast cancer screening, while follow-up mammography may increase the diagnostic rate when compared to the prior mammography. We suggest that the public health authority can encourage subjects to undergo screenings in the same health institute where they regularly visit.

## 1. Introduction

Breast cancer accounts for 30% of all new cancer diagnoses for women and is a heterogeneous disease presenting various clinical behaviors, morphological appearances, and treatment responses [1]. The highest percentage of breast cancer occurs as invasive ductal carcinoma, accounting for 85% to 90%. There are many factors influencing the clinical prognosis of breast cancer patients, including patients’ clinical conditions (i.e., age, tumor size, histological grading, status of lymph nodes, and menopausal status), the expression of estrogen and progesterone receptors, and the expression of human epidermal growth factor receptor 2 (c-erbB-2) [2]. Breast cancer is the second leading cause of cancer death in women of all ages in America and the fifth leading cause of death worldwide [3]. In Taiwan, breast cancer is the third leading cause of cancer death in women [4]. Its mortality rate remains high, although there have been improvements in the screening techniques.

Early detection in the clinical setting plays an important role in cancer treatment. The goal of screening is to find cancers when they are still curable (i.e., non-palpable and without regional node metastases) to decrease cancer-related mortality. Regular mammography screening allows the early detection of benign and malignant tumors, thus facilitating immediate treatment and significantly reducing the mortality rate and treatment-related morbidity [5,6]. Early detection with screening mammography significantly reduces breast cancer deaths by 20% to 40% [7,8,9,10,11,12,13]. Annual mammography screening for women between the ages of 40 to 84 prevents more cancer deaths than biennial screening for women aged 50 to 74 years [14]. Tabár et al. qualitatively concluded that a substantial and significant reduction in breast cancer mortality was associated with an invitation for screening after three-decade follow-up [15,16].

Although mammography screening is widely considered effective, the distress associated with false positive results cannot be overlooked. For women who started receiving screening mammography annually at age 40, the 10-year cumulative probability of a false positive recall is 61% in Taiwan [17]. In Japan, the neighboring country of Taiwan, a previous study noted that 59.3% of cases were categorized as having dense breast tissues where the sensitivity of mammography or ultrasonography alone did not exceed 80% [18]. On average, when a woman starts the annual screening between 40 to 49 years of age, she may encounter one false positive mammogram per 10 years, and one in four women will receive one false positive biopsy [15]. Although the false positive rate is relatively high, such annual screening mammography does reduce the mortality among women aged 40–84 [19]. Due to the reduced frequency of performed imaging, biennial screening can reduce the false positivity of mammograms by one-third in comparison with annual screening [17,20].

Previous studies focused on the fact that regular screening can reduce mortality. Our study focuses on the theory that regular screening within the same institution can increase the true positive rate, reduce patient anxiety, facilitate medical management, and avoid unnecessary medical expenses. The aim of this study was to compare the performance of screening mammography based on first visits and follow-up visits and to determine the effect of prior mammograms on screening performance, recall rates, and positive predictive values (PPVs).

## 2. Materials and Methods

### 2.1. Screening Population

In Taiwan, the peak age for breast cancer diagnosis is between 45 and 69 years old. Therefore, the Health Promotion Administration offers a free mammography exam biennially for women aged 45 to 69 and for those aged 40 to 44 years with a family history of breast cancer (paternal and maternal grandmothers, mothers, daughters, and sisters who have had breast cancer).

The research sample of this study consisted of women who received a free mammography exam offered by the Health Promotion Administration in Taiwan from 1 January 2015 to 31 December 2018. The study design was approved by the Institutional Review Board in our hospital (KMUHIRB-E(I)-20220019).

Women with existing clinical symptoms (e.g., a palpable breast mass) were excluded. Women who had been diagnosed with breast cancer more than 10 years prior to this study and were previously declared disease-free by their physicians were enrolled. The examinees’ clinical data (education level, personal disease history, family history of breast cancer, menopausal status, childbearing history, history of breast surgery, and whether abnormal masses are identified) were recorded by the technician who conducted the mammography. The examinees filled out a written informed consent form to allow their data to be transferred to the screening database.

A total of 12,301 women participated in this study, and 15,661 mammography screenings were performed. Of them, 626 screenings were visits due to clinical symptoms and were excluded. Of the remaining 14,935 screenings (Figure 1), 10,040 screenings (67.2%) were first visits and 4895 screenings (32.8%) were follow-up visits.

As the Health Promotion Administration offers a free mammography exam biennially, this study included 4 years’ worth of data. Thus, if a woman was regularly screened at this hospital, the same woman would have 2 screening data records.

### 2.2. Mammography Imaging

The mammography images were taken with a Giotto Image 3D full-field digital mammography unit (IMS Giotto, Sasso Marconi, Italy). A movable grid was used with voltage ranging from 22 to 35 kVp to obtain the mammograms. The mAs value is an important parameter indicating the radiation dose to patients, and it ranges from 4 to 500 mAs, with an interval of no more than 15%. In our study, it was set to around 30 to 100 mAs. The mammography method involved the standard bilateral craniocaudal view and mediolateral oblique view. Each woman received a total of 4 mammograms. The physician’s screen for image interpretation was a display with a resolution of 5M pixels. All mammograms were interpreted by the radiologists. The CDR was obtained by dividing the number of participants diagnosed with breast cancer by the total number of participants screened (CDR per 1000 screenings).

### 2.3. Screening

The screened women were divided according to whether it was their first visit or a follow-up visit for breast cancer screening. First visit means that the participant came to this hospital for examination for the first time. This could mean that the participant met the qualifications of the Health Promotion Administration for the first time and came to this hospital for screening (first screening), or that the participant, who had been screened at a different location before, visited this hospital for the first time for screening (previous images were unavailable). Follow-up visit means that the participant had been screened in this hospital before, possessed previous imaging data, and was also screened in this hospital this time (with previous images). The mammography report was described using the 5th edition of the Breast Imaging Reporting and Database System score (BI-RADS) of the American College of Radiology (ACR).

Categories 1 to 3 were classified as negative; Categories 4, 5, and 0 were classified as positive. Participants with Category 0 were requested to return for a follow-up to undergo other examinations and reassessment. If examinees were classified as Category 1, 2, or 3, regular examination for them was recommended. If the follow-up results were Category 4 or 5, surgical tests, namely fine-needle aspiration biopsy, pathological tissue biopsy, core biopsy, hook wire needle localization, stereotactic localization, and surgical biopsy, were recommended. The pathological results were obtained from the screening database of the Ministry of Health and Welfare and were divided into benign and malignant tumors. Benign breast diseases were defined as non-breast cancer, whereas ductal carcinoma in situ (DCIS) and invasive carcinoma were defined as breast cancer.

### 2.4. Statistical Analysis

Descriptive statistics were adopted to classify the data and acquire the percentages (%). The specificity, PPV, cancer detection rate (CDR), and recall rate were used to evaluate and analyze the overall screening performance. SPSS 12.0 version was used for all statistical analyses.

## 3. Results

### 3.1. Patient Demographics

This study consisted of 11,892 participants, of whom 3043 (25.6%) participated in two screenings in this study and 8849 (74.4%) participated in one screening. The mean age of the women undergoing screening was 54 years. The percentages of the age groups at screening were as follows: 5.3% were screened at the age of 40 to 44 years, 26.7% were screened at the age of 45 to 49 years, 23.4% were screened at the age of 50 to 54 years, 20.9% were screened at the age of 55 to 59 years, 15.3% were screened at the age of 60 to 64 years, and 8.5% were screened at the age of 65 to 69 years. In terms of education level, most participants graduated from high school/vocational high school (48.5%), followed by those who attended junior college/university (30.1%), junior high school (10.0%), elementary school (7.4%), or graduate school or higher (3.2%) and those with no formal education (0.9%). Most women had no history of breast cancer (73.4%), followed by those with benign breast tumor (25.2%), those with other cancers (1.1%), and those who previously had breast cancer (0.2%). Participants with and without a history of breast cancer within their second-degree relatives accounted for 73.4% and 6.3%, respectively. Furthermore, 88.4% had a childbearing history, 60.1% were menopausal, and 91.0% had no history of breast surgery. In terms of the classification of the participants’ mammary glands, most were classified as heterogeneous (68.1%), followed by extremely dense (18.5%), scattered fibroglandular (13.0%), and fatty (0.4%).

### 3.2. Positive and Negative Results

The initial mammography interpretation results are listed in Table 1. The proportion of negative screening results for women who returned for follow-up visits (96.0%) was higher than that for first-visit participants (90.7%, *p* < 0.0001). In comparison, the proportion of positive screening results for first-visit participants (9.3%) was higher than that for their follow-up counterparts (4.0%, *p* < 0.0001).

### 3.3. Biopsy and Pathological Confirmation

Table 2 presents the results that were classified as positive (BI-RADS Categories 0, 4, and 5) and the subsequent surgical biopsy results. First-visit participants had a higher number of suspicious lesions that were determined to be benign (73.1%) than follow-up visit participants (64.2%) and a lower number of results that were determined to be malignant (26.9%) than follow-up participants (35.8%, *p* = 0.1491).

A total of 98 participants (74 first-visit participants and 24 follow-up participants) were confirmed to have breast cancer in this study. The pathology report concluded a slight difference between the percentage of DCIS among first-visit (46.0%) and follow-up visit participants (41.7%). In comparison, a higher percentage of follow-up visit participants (58.3%) were diagnosed with invasive carcinoma compared to first-visit participants (50.0%).

Most breast cancer cases detected by screening had no lymphatic metastasis. Moreover, little difference was observed between the two groups of participants in terms of whether cancer metastasis was observed (81.1% and 87.5% of first-visit and follow-up visit participants, respectively, had no axillary lymph node metastasis, and 10.8% and 12.5% of first-visit and follow-up visit participants, respectively, indicated cancer metastasis). The maximum size of the breast cancer masses detected among follow-up visit participants was <20.0 mm (mean size: 10.1 mm). However, 24.3% of first-visit participants had masses >20.0 mm (mean size: 16.8 mm).

### 3.4. Performance of Mammography Screening

The effect of comparing first-visit and follow-up visit participants on the performance and effectiveness of mammography is described as follows and in Table 3.

#### 3.4.1. Recall Rate

Table 3 presents the recall rate (the proportion of positive mammography results, i.e., BI-RADS Categories 0, 4, and 5). The overall recall rate was 7.9%, and first-visit participants had a higher recall rate (9.7%) than their follow-up visit counterparts (4.2%, *p* < 0.0001).

#### 3.4.2. CDR

The overall CDR was 6.6 per 1000. As presented in Table 3, first-visit participants had a higher CDR (7.4 per 1000) than follow-up visit participants (4.9 per 1000, *p* = 0.0761).

#### 3.4.3. PPV

PPV was divided into PPV1, PPV2, and PPV3, among which PPV1 was the number of participants with positive mammography results (BI-RADS Categories 0, 4, and 5) who were actually diagnosed with breast cancer. The overall PPV1 (abnormal detection in screening) was 8.3%, and the PPV1 value of follow-up visit participants (11.8%) was significantly higher than that of first-visit participants (7.6%, *p* < 0.0001).

PPV2 (biopsy recommended) was the number of participants with positive mammography results (BI-RADS Categories 4 and 5) who were actually diagnosed with breast cancer. The overall PPV2 was 25.7%, and the PPV2 value of follow-up visit participants (32.0%) was significantly higher than that of first-visit participants (24.2%, *p* < 0.0001).

PPV3 (biopsy conducted) was the number of participants with positive mammography results (BI-RADS Categories 4 and 5) who underwent biopsy tests (true positive + false positive) and were actually diagnosed with cancer. The overall PPV3 was 28.7%, and the PPV3 value of follow-up visit participants (35.8%) was significantly higher than that of first-visit patients (27.0%, *p* < 0.0001).

## 4. Discussion

In the screening of breast cancer, the recall rate is one of the quality assurances to evaluate the performance of radiologists. The recall rate should be low without compromising the CDR due to the increases in clinical cost and patients’ anxiety associated with callbacks after screening mammograms [21]. The European guidelines recommend a target recall rate of 5% [22], whereas the ACR recommends the range of 5% to 12% to maximize sensitivity and specificity [23]. In the current study, the overall recall rate for the examinees was 7.9%, which is a little higher than the recommendation of the European guidelines (desirable standard of <5% for first visit and <3% for follow-up) [22] but within the range of the ACR recommendation (5–12%) [23]. When previous images were available for comparison (follow-up visits), the recall rate was significantly improved (4.2%), with the follow-up visit and first-visit participants displaying an approximately 2.3-fold difference (9.7% first visit vs. 4.2% follow-up).

As seen from the results, the CDR was not as favorable as expected at the beginning of the study, with fewer cases of breast cancer detected in the follow-up visit participants (4.9 per 1000) than in the first-visit (7.4 per 1000) counterparts. Although the CDR in the follow-up visit group failed to be higher than that of the first-visit group in the current study, the overall CDR was still superior to the data in other studies. Previous reports showed that the detection rate of screening mammography was 2 to 8 cancers per 1000 mammograms [23,24,25].

In the current study, the overall PPV3 for examinees was 28.7%. The performance was similar to that in the literature review [14]. Moreover, in the current study, the PPV3 for the first-visit and follow-up visit groups was 27.0% and 35.8%, respectively. It revealed that the availability of previous mammography data for the physician’s reference improved the mammography screening performance. Follow-up visits—i.e., where previous mammograms were available for the doctor’s reference—yielded more favorable results in their data.

As to the size of an early breast cancer detected by the screening, when physicians had access to previous imaging data for reference (follow-up visits), the mass size of the breast cancer detected did not exceed 2 cm. However, when the physicians did not have access to previous imaging data (first visits), a proportion of the patients diagnosed with breast cancer did have masses exceeding 2 cm. Moreover, the data indicated that when physicians had access to previous images (follow-up visits), breast cancer could be detected immediately when the mean mass size was only 1 cm. If no image was available for comparison (first visits), breast cancer could only be detected when the mass grew to a mean of 2 cm.

The sensitivity of mammography is lower in women with dense breasts [26,27,28,29]. The reason is the similarity of X-ray attenuation between cancers and the dense fibroglandular tissue, in which obscuration of the tumor occurs [30]. With this limitation, supplemental screening modalities, such as digital breast tomosynthesis (DBT), screening ultrasound, and screening magnetic resonance imaging (MRI), have been investigated. With the supplemental screening modalities, the performance of screening may be influenced. For example, one study showed an average recall rate of 7.9% (range from 5.5% to 9.5%) after applying DBT and 6.8% (range from 3.6% to 9.7%) before DBT (*p* = 0.0316) [31]. Compared to the current study, the recall rate after DBT is similar to that of our overall value; however, it is higher than that of our follow-up visit group.

This study has provided clinical evidence of the effects of screening mammography, especially at follow-up visit. However, it still has the following limitations. Among the positive cases in this study, not every screened participant returned for follow-up or biopsy examinations, and these data would affect the research results. In addition, mammogram interpretation was conducted by two radiologists, and the inter-observer bias of interpretation needs to be further evaluated.

Accordingly, when performing large-scale screening mammography, we encourage the public to undergo mammography screening at the same health institute, which is helpful to the overall performance of mammography and can reduce the possibility of false positive results. The government and health units are recommended to provide relevant information to the public as a reference. According to prior data, about 15% of breast cancer diagnoses may not be detected by mammography, depending on technical limitations and/or operator skill. If patients undergo mammography screening at the same health institute, it may prevent false positive results. A follow-up mammography may increase the PPV of disease diagnosis as compared with prior mammography. On the other hand, patients are encouraged to have a digital copy of each mammography as their own record, so they can bring them for the next mammography. The authors hope that the settings for viewing images on the cloud system will be more comprehensive in the future to provide references for physicians and facilitate the public’s convenience.

## 5. Conclusions

Mammography can detect the transformation of breast microcalcifications and abnormal shadows of early lesions along with early detection of malignant tumors. If prior mammography images are available to be compared, the PPV can be increased and the recall rate can be reduced. In this study, the average lump size, if confirmed to be malignant, was within 1 cm. The public health authority should encourage subjects to undergo screenings at the same health institute where they regularly visit. In our study, when the previous mammography images could be compared, the PPV increased and the recall rate was reduced. This may establish a system for early detection and prevention.

## Figures and Tables

**Figure 1 healthcare-10-01037-f001:**
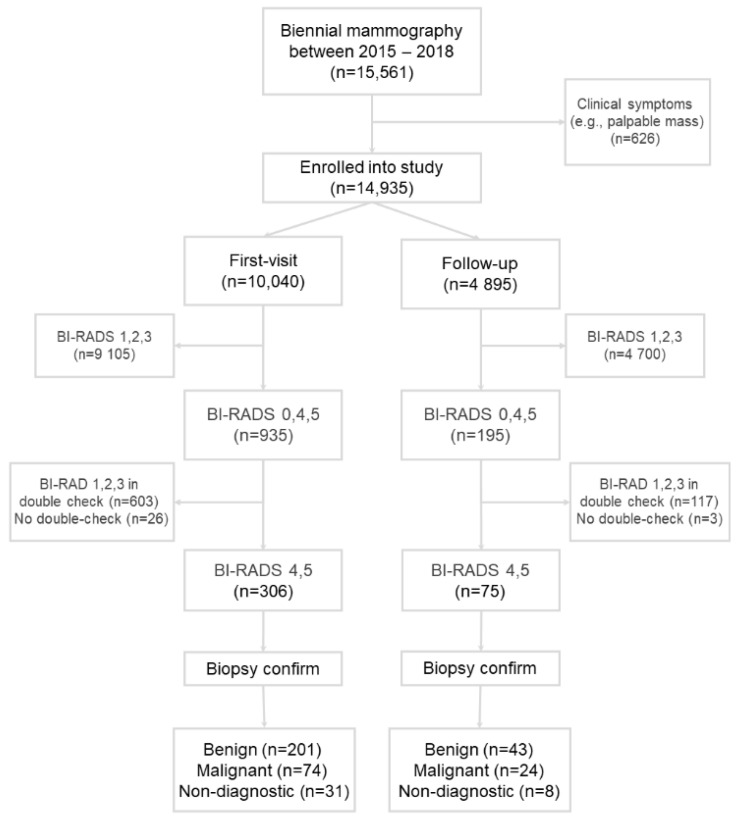
Flowchart of the study.

**Table 1 healthcare-10-01037-t001:** Initial interpretation results of mammography images.

		First-Visit		Follow-Up		*p*-Value
	BI-RADS	Mammography	%	Mammography	%	
Negative	1	1343	90.7	586	96.0	<0.0001
	2	7726		4105		
	3	36		9		
Positive	4	35	9.3	11	4.0	<0.0001
	5	2		0		
	0	898		184		
Total		10,040	100	4895	100	

**Table 2 healthcare-10-01037-t002:** The surgical biopsy results of examinees with positive mammography findings.

	First-Visit		Follow-Up		*p*-Value
	Mammography	%	Mammography	%	
Benign	201	73.1	43	64.2	0.1491
Malignant	74	26.9	24	35.8	
Total	275	100	67	100	

**Table 3 healthcare-10-01037-t003:** Performance of the mammography screening.

	First-Visit + Follow-Up	First-Visit	Follow-Up	*p*-Value
Recall rate (%)	7.9	9.7	4.2	<0.0001
CDR (per 1000)	6.6	7.4	4.9	0.0761
PPV1 (%)	8.3	7.6	11.8	<0.0001
PPV2 (%)	25.7	24.2	32.0	<0.0001
PPV3 (%)	28.7	27.0	35.8	<0.0001

## Data Availability

The data presented in this study are available on request from the corresponding author.

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
