# Peer review of "The Effects of Prior Mammography Screening on the Performance of Breast Cancer Detection in Taiwan"

_healthcare, 2022, doi:10.3390/healthcare10061037_

Round 1

Reviewer 1 Report

Interesting study, with a great number of patients, mainly in the first visit group which can biased final results. Also, the real impact of a screening program will be seen at least after 14 years (Tabar and colls). is expected to have more cancer detected in first rounds of a screening program, also a higher biopsy rate and recalls. Not sure that the results can be trusted with this study design. 

more detail commnets:
My opinion and main concern about the paper Title: The Effects of Prior Mammography Screening on the Performance of breast cancer detection, is based on the study design since for me is not fair to compare first round of screening vs follow up, in the way the authors did, the results are expected, it is well known that previous studies improve diagnostic. Also, more cancers are found in first study round, depending age group and country.

The effect of breast cancer screening on population-based breast cancer mortality does not appear immediately but takes years to accumulate. there is evidence that there are clear benefits of repeated regular screening.  

1. Tabàr L, Fagerberg G, Duffy SW, Day NE, Gad A, Gröntoft O. Update of the Swedish two-county program of mammographic screening for breast cancer. Radiol Clin North Am 1992;30(1):187–210. Medline, Google Scholar

2. Tabár L, Vitak B, Chen TH, et al. Swedish two-county trial: impact of mammographic screening on breast cancer mortality during 3 decades. Radiology 2011;260(3):658–663. Link, Google Scholar

Author Response

Interesting study, with a great number of patients, mainly in the first visit group which can biased final results. Also, the real impact of a screening program will be seen at least after 14 years (Tabar and colls). is expected to have more cancer detected in first rounds of a screening program, also a higher biopsy rate and recalls. Not sure that the results can be trusted with this study design. 

more detail commnets:
My opinion and main concern about the paper Title: The Effects of Prior Mammography Screening on the Performance of breast cancer detection, is based on the study design since for me is not fair to compare first round of screening vs follow up, in the way the authors did, the results are expected, it is well known that previous studies improve diagnostic. Also, more cancers are found in first study round, depending age group and country.

The effect of breast cancer screening on population-based breast cancer mortality does not appear immediately but takes years to accumulate. there is evidence that there are clear benefits of repeated regular screening.  

  1. Tabàr L, Fagerberg G, Duffy SW, Day NE, Gad A, Gröntoft O. Update of the Swedish two-county program of mammographic screening for breast cancer. Radiol Clin North Am 1992;30(1):187–210. Medline, Google Scholar
  2. Tabár L, Vitak B, Chen TH, et al. Swedish two-county trial: impact of mammographic screening on breast cancer mortality during 3 decades. Radiology 2011;260(3):658–663. Link, Google Scholar

Reply: The authors appreciated that the reviewer pointed out this comment. We modified the title as "The Effects of Prior Mammography Screening on the Performance of Breast Cancer Detection in Taiwan" to highlight the differences in breast cancer screening across countries. In our manuscript, references 6-13 are all about the impact of screening on cancer mortality. These two references were added and cited as ref. 14 and 15.

We also added the sentences in the revised manuscript. "Previous studies focused on the fact that regular screening can reduce mortality. Our study is focusing on the theory that regular screening within the same institution can increase the true positive rate, reduce patient anxiety, facilitate medical management and avoid unnecessary medical expense."

Reviewer 2 Report

This is an interesting and important study. I have these suggestions:

p2l50 and l65 add the statistics for Taiwan, are they different from prior studies in other countries or generally similar

p2l87 add 10 years prior to this study

p3 l126 change specifically to specificity

l132 should be included

p3 through p6 all the results should separate first mmg ever vs first at HPA but had prior mmg elsewhere vs follow up mmg at HPA after prior mmg at HPA, and all the results should have p values for differences (or no differences) between those groups. This will help make these results better defined and more important. Fig 1 would include these analyses and split first visit column to first ever and first at HPA but follow up after prior non HPA mmg. 

p5 l180 add actual stage results based on pathology for each groups

p6 l228 show these data in results section and show p value

p7 l246 show these data in results, p value for any difference of radiologist having prior films vs not having prior films

p7l258 change the study to a study, and compare those results with this current study

p7l267 this applies to both large scale screening mammography and to any mammography even in individual patients. Emphasize the possible importance of having a digital copy of each mmg in the patients own records so she can bring to the next mmg (since so many patients change health plans or change where mmg is done due to contracting changes)

results: were there differences in mmgs done by non tomographic methods vs those done with tomosynthesis? give statistical comparison and p value 

Author Response

This is an interesting and important study. I have these suggestions:

p2l50 and l65 add the statistics for Taiwan, are they different from prior studies in other countries or generally similar

Reply: The authors appreciated that the reviewer pointed out this comment. We revised and added new references in our manuscript.

Breast cancer is the second leading cause of cancer death in women in American and the fifth leading cause of death worldwide [3]. In Taiwan, breast cancer is the third leading cause of cancer death in women [4].

In Japan, the neighboring country of Taiwan, a previous study noted that 59.3% of cases were categorized as having dense breast tissues where the sensitivity of mammography or ultrasonography alone did not exceed 80% [18].

p2l87 add 10 years prior to this study

Reply: The authors appreciated that the reviewer pointed out this comment. It was revised.

p3 l126 change specifically to specificity

Reply: The authors appreciated that the reviewer pointed out this comment. It was revised.

l132 should be included

Reply: The authors appreciated that the reviewer pointed out this comment. In this study, "excluded" is correct, not "included".

p3 through p6 all the results should separate first mmg ever vs first at HPA but had prior mmg elsewhere vs follow up mmg at HPA after prior mmg at HPA, and all the results should have p values for differences (or no differences) between those groups. This will help make these results better defined and more important. Fig 1 would include these analyses and split first visit column to first ever and first at HPA but follow up after prior non HPA mmg. 

Reply: The authors appreciated that the reviewer pointed out this comment. Reviewer recommends dividing the patients into three groups: first mmg ever vs first at HPA but had prior mmg elsewhere vs follow up mmg at HPA after prior mmg at HPA. However, we checked the original data in our records and found out that the patients were divided into two groups: the first mammography (the first group), and the non-first time (the second plus the third group). The second and third groups cannot be subdivided. This is a valuable suggestion, and we will pay attention to the grouping of patients in our future studies.

p5 l180 add actual stage results based on pathology for each groups

Reply: The authors appreciated that the reviewer pointed out this comment. Due to the mammography is only used for preliminary imaging diagnosis, even if the subsequent biopsy is confirmed to be malignant, the patient will be transferred to a medical center for treatment. Before surgery, other examination such as computed tomography is required to know the stage of the cancer (including the size of the tumor, the presence or absence of lymph node invasion, and the distal transfer). The information of CT with pathology reports were kept in different medical center. Our IRB can only access those patient records at our hospital, not records from other medical centers. Therefore, we cannot provide pathology reports or tumor stages of the patients.

p6 l228 show these data in results section and show p value

Reply: The authors appreciated that the reviewer pointed out this comment. The p value was shown in Table 3 in the results section (red marker).

p7 l246 show these data in results, p value for any difference of radiologist having prior films vs not having prior films

Reply: The authors appreciated that the reviewer pointed out this comment. The p value was shown in Table 1 and 3 and also in the results section.

p7l258 change the study to a study, and compare those results with this current study

Reply: The authors appreciated that the reviewer pointed out this comment.

We changed the study to a study. Compared to the current study, the recall rate after DBT is similar to that of our overall value; however, it is higher than that of our follow-up visit.

p7l267 this applies to both large scale screening mammography and to any mammography even in individual patients. Emphasize the possible importance of having a digital copy of each mmg in the patients own records so she can bring to the next mmg (since so many patients change health plans or change where mmg is done due to contracting changes)

Reply: The authors appreciated that the reviewer pointed out this comment. We added those sentences in the discussion section.

"According to prior data, it may have about 15% breast cancer that could not be detected by mammography. It depends on technical limitation or operation skill. If patients undergo mammography screening at the same health institute, it may prevent the false positive result. A follow-up mammography may increase PPV of disease diagnosis as compared with prior mammography. On the other hand, patients are encouraged to have a digital copy of each mammography as the patients’ own records, so they can bring them for the next mammography."

results: were there differences in mmgs done by non tomographic methods vs those done with tomosynthesis? give statistical comparison and p value 

Reply: The authors appreciated that the reviewer pointed out this comment.

In our study, MMGs were imaged by non-tomographic methods. Breast tomosynthesis is another imaging modality for breast cancer screening; however, not every patient undergoes tomosynthesis. Therefore, it may be difficult to compare with tomosynthesis.

Reviewer 3 Report

Review:

Chang et al.: The Effects of Prior Mammography Screening on the Performance of Breast Cancer Detection

General remarks:

The Paper is well written and structured/designed. However, I have a few comments to improve the manuscript. The comments are specified as follows:

Abstract:

The abstract is also well written, but it seems to be missing overall conclusion sentences at the end of the abstract.

Introduction:  

Well written and structured with good citations from previously conducted studies. Thus, no remarks.

Methods:

The kVp is a crucial parameter in accordance with the breast tissue thicknesses. Thus, the kVp ranges give an indication of the ranges of the breast tissue thicknesses of the wholes study. In line 96 the author mentioned only the lower limit of the kVp, which is 22 kVp. Therefore, it would be good to state the kVp range with lower and upper limits or average values.

Another comment regarding exposure parameters is that not all other important parameters such as mAs and filter-target combinations have been mentioned.

The mAs value is an important parameter indicating the radiation dose to patients; therefore, I would like to suggest that mAs ranges should be stated/notified.

Results:

Lines 130 to 156 of the Results section including a flow chart of the study sound like a description of methods. The authors might consider moving this section to the method section.

Author Response

Chang et al.: The Effects of Prior Mammography Screening on the Performance of Breast Cancer Detection

General remarks:

The Paper is well written and structured/designed. However, I have a few comments to improve the manuscript. The comments are specified as follows:

Abstract:

The abstract is also well written, but it seems to be missing overall conclusion sentences at the end of the abstract.

Reply: The authors appreciated that the reviewer pointed out this comment. We added two overall conclusion sentences at the end of the abstract as below.

By this study, we concluded that a prior mammography played an important role for breast cancer screening while the follow-up mammography may increase the diagnostic rate when compared to the prior mammography. We suggest that the public health authority can encourage subjects to conduct screenings in the same health institute where they regularly visit.

Introduction:  

Well written and structured with good citations from previously conducted studies. Thus, no remarks.

Reply: The authors appreciated the reviewer for their recognition and encouragement.

Methods:

The kVp is a crucial parameter in accordance with the breast tissue thicknesses. Thus, the kVp ranges give an indication of the ranges of the breast tissue thicknesses of the wholes study. In line 96 the author mentioned only the lower limit of the kVp, which is 22 kVp. Therefore, it would be good to state the kVp range with lower and upper limits or average values.

Reply: The authors appreciated that the reviewer pointed out this comment. The kVp was ranged from 22 to 35 kVp. It was revised in the manuscript.

Another comment regarding exposure parameters is that not all other important parameters such as mAs and filter-target combinations have been mentioned.

The mAs value is an important parameter indicating the radiation dose to patients; therefore, I would like to suggest that mAs ranges should be stated/notified.

Reply: The authors appreciated that the reviewer pointed out this comment. "The mAs value is an important parameter indicating the radiation dose to patients. It was ranged from 4 to 500 mAs, with interval no more than 15%. In our study, it was set around 30 to 100 mAs." It was added in the manuscript.

Results:

Lines 130 to 156 of the Results section including a flow chart of the study sound like a description of methods. The authors might consider moving this section to the method section.

Reply: The authors appreciated that the reviewer pointed out this comment. Lines 130 to 156 of the Results section including a flow chart were moved to the method section.